# OpenReview forum: "Attacking Gray-Box Large Vision-Language Models with Adaptive SVD-Structured Adversarial Alignment"
_ICML.cc/2026/Conference — ICML 2026 regular_

### Official Review · Reviewer_e59t · 2026-03-06

**Soundness:** 4
**Presentation:** 3
**Significance:** 4
**Originality:** 3
**Overall Recommendation:** 5
**Confidence:** 4

**Summary:**

This paper investigates a critical issue of gray-box adversarial attacks against LVLMs, by introducing a novel perspective of SVD-structured adversarial alignment. The core insight is that previous works rely on specific selected target images to adjust the adversarial feature while ignoring the rich and multi-view adversarial alignment between the detailed visual and target semantics. Therefore, it proposes to perturb the visual feature to best match more natural attacker-chosen target texts, and develop a global-local semantic alignment module via a SVD-structured subspace for learning optimal transport. Extensive experiments are validated across various LVLM models with multiple LVLM datasets/tasks compared with existing LVLM attacks.

**Compliance With Llm Reviewing Policy:**

Affirmed.

**Key Questions For Authors:**

1. While the proposed method is effective, it is still relatively expensive: it requires a "augmentation" step with an LLM, estimation of the principal components, computation and projection of multiple input features, and finally, the optimal transport algorithm itself. Admittedly, the first two steps can be cached, but the projection and optimal transport must be computed for all inputs. More discussion is required.

2. The OT framework introduces its own set of sensitivities. Further, the weights assigned to the discrete distributions are determined by a heuristic based on prediction entropy. It is also unclear why entropy is a good measure to estimate the importance weights.

3. The quality of the semantic subspace is fundamental to the method. How sensitive is the performance to the richness of the text prompts?

4. There are various existing LVLM attacks are re-implemented for comparison. Do they re-implemented in the same setting for fair comparisons? More details are required.

5. Target text selection. Since this work proposes a first target semantic guidance for gray-box attack, it would be valuable to include experiments with more specific target texts and report the success rates for such cases.

**Limitations:**

Limitations and potential negative impacts have been sufficiently discussed in the paper.

**Strengths And Weaknesses:**

**Strengths**

This paper addresses a significant gap in the field of adversarial attacks on LVLMs, offering a more practical and versatile approach compared to existing gray-box LVLM attack methods. The SVD-guided adaptive feature adversarial alignment and its ability make it particularly relevant for real-world applications and raise important questions about the security of LVLMs.

1. The technical approach presented is novel and well-motivated. It proposes a target text adversarial alignment with novel SVD-structured global-local feature learning for handling the complicated gray-box LVLM attacks. I think it is meaningful for the LVLM adversarial field.

2. Theoretical analyses are provided to support the proposed framework. The illustrated attack process is clear and easy to follow.

3. Sufficient experiments provided in both the main paper and appendix validate the effectiveness of the proposed method.

4. The safety concerns of LVLMs exposed in this paper could potentially have a large impact on the AI safety field.

**Weaknesses**

The reviewer finds that this paper is novel and provides valuable insights to the community. However, there are several areas where the paper should be addressed:

1. While the proposed method is effective, it is still relatively expensive: it requires a "augmentation" step with an LLM, estimation of the principal components, computation and projection of multiple input features, and finally, the optimal transport algorithm itself. Admittedly, the first two steps can be cached, but the projection and optimal transport must be computed for all inputs. More discussion is required.

2. The OT framework introduces its own set of sensitivities. Further, the weights assigned to the discrete distributions are determined by a heuristic based on prediction entropy. It is also unclear why entropy is a good measure to estimate the importance weights.

3. The quality of the semantic subspace is fundamental to the method. How sensitive is the performance to the richness of the text prompts?

4. There are various existing LVLM attacks are re-implemented for comparison. Do they re-implemented in the same setting for fair comparisons? More details are required.

5. Target text selection. Since this work proposes a first target semantic guidance for gray-box attack, it would be valuable to include experiments with more specific target texts and report the success rates for such cases.

---

> ### Author Rebuttal · Authors · 2026-03-30
>
> # Response to Reviewer e59t
>
> **Q1:** Computational cost for projection and optimal transport.
>
> **A1:** Thanks for the comment. First of all, we want to clarify that these operations do not introduce much more time costs. We have provided detailed comparisons of computational costs in Table 4 of the paper, where our method is competitively efficient in both FLOPs and Time Costs. Second, the "augmentation" step with an LLM and estimation of the principal components only require one computation for the same target text. The projection step involves simple matrix multiplication between a $d$-dimensional input feature and the prompt-specific projection matrix, with a per-input cost of $\mathcal{O}(d^2)$, which is negligible in practice (e.g., <0.08s per sample). Third, for OT, its complexity is generally $\mathcal{O}(M \cdot N \cdot \log(\frac{1}{\tau}))$, where $M$ and $N$ denote the number of augmented views and text descriptions, $\tau$ is the regularization coefficient and usually set to 0.01. Importantly, this OT computation is performed in parallel across all prompts, resulting in efficient runtime. We will add more discussion.
>
> **Q2:** Concerns over entropy-based weighting.
>
> **A2:** Thanks for your concern. We agree that the OT framework introduces its own set of sensitivities, and the misclassified views may receive lower entropy and thus higher weights. To examine this concern, we conducted an empirical analysis on the COCO dataset and found that the average weight assigned to such views is only 0.24, indicating that while these cases exist, they are non-dominant in the overall distribution. More importantly, the OT solver computes a global transport plan across the entire set, guided by the pairwise similarity structure between image and text distributions. Therefore, the few biased views cannot dominate the result due to OT’s normalization constraints.
>
> Besides, entropy has been widely adopted in test-time adaptation and prompt tuning [1,2], as it directly reflects model confidence by quantifying output certainty. To assess its impact, we have compared entropy-based weighting with uniform weighting. As shown below, entropy-based weights consistently yield better robustness across datasets. Therefore, we choose entropy to estimate the importance weights.
>
> |Method|Qwen2.5-VL-3B (ASR)|Qwen2.5-VL-3B (AvgSim)|LLaVa-1.5-7B (ASR)|LLaVa-1.5-7B (AvgSim)|
> |:----:|:----:|:----:|:----:|:----:|
> |entropy-based|86.5|0.82|84.7|0.79|
> |uniform weights|84.3|0.74|81.9|0.68|
>
> [1] Test-Time Prompt Tuning for Zero-Shot Generalization in Vision-Language Models, 2022
>
> [2] Diverse Data Augmentation with Diffusions for Effective Test-time Prompt Tuning, 2023
>
> **Q3:** Impact of different prompt number.
>
> **A3:** In fact, as for semantic subspace, SVD-structured projection is designed to extract the most dominant principal components (Top-$K$) from the combined semantic contexts, so the subspace projection step is relatively insensitive to prompt richness. However, OT operates on distributions, when only a few prompts are used, the textual side collapses to a point mass, and the OT objective no longer captures meaningful distributional alignment. Consequently, the transport plan becomes degenerate and fails to effectively model semantic matching across multiple views. We have conducted the ablation study using different prompt number $N$ (Local Target Number) in Figure 5 of the paper, where increasing $N$ consistently improves performance, and we set $N=30$ according to this empirical study.
>
> **Q4:** Implementations of previous works.
>
> **A4:** Thanks for your concern. Our implementations include two parts: (1) Gray-box attack in Table 1 of the paper, where we carefully re-implement previous attack methods into our same setting for fair comparisons, using the same vision encoder, the same budget $\epsilon=4/255$, the same step size of 1/255 and the same epoch of 100 iterations. (2) CLIP-based transfer attack in Table 2 of the paper, where we carefully implement our attack into the same setting as the compared methods for fair comparisons, using the same vision encoder, the same budget $\epsilon=16/255$, the same step size of 1/255 and the same epoch of 300 iterations. Overall, our comparisons with existing LVLM attacks are fair. We will add more clarification in the revision.
>
> **Q5:** More target texts.
>
> **A5:** We provide the experiments on the specific target texts and report the attack success rates in the table below. It shows that our attack is insensitive to the target text. We will add them in the revision.
>
> |Target Text|Qwen2.5-VL-3B|Qwen2.5-VL-7B|LLaVa-1.5-7B|LLaVa-1.6-7B|Gemma-3-4B|Gemma-3-12B|
> |:----:|:----:|:----:|:----:|:----:|:----:|:----:|
> |"A man holding a big doughnut at a festival"|87.2|82.3|85.6|79.1|83.8|75.0|
> |"A photo of a teddy bear on a skateboard in Times Square"|84.5|79.6|82.8|76.4|81.2|72.3|
> |"A beautiful bird with a black and white color in snow"|88.1|83.5|86.9|80.2|84.6|76.8|

---

> > ### Author Rebuttal · Reviewer_e59t · 2026-04-02
> >
> > The authors have addressed all my concerns with quantitative evidence and clear explanations. After seeing the comments of other reviewers and the rebuttals, I believe this work brings novel insight of target text adversarial alignment for practical gray-box LVLM attack with SVD-structured global-local feature learning. The proposed method also achieves impressive performances. Based on these points, I maintain my score of 5 and vote for acceptance.

---

> > > ### Author Response · Authors · 2026-04-02
> > >
> > > We would like to thank the reviewer for responding to our rebuttal. It is great to know that your concerns have been addressed. We will carefully incorporate the rebuttal in detail into the revised version.

---

### Official Review · Reviewer_kPjo · 2026-03-11

**Soundness:** 2
**Presentation:** 3
**Significance:** 3
**Originality:** 2
**Overall Recommendation:** 4
**Confidence:** 4

**Summary:**

This paper proposes a gray-box adversarial attack on Large Vision-Language Models (LVLMs) that requires access only to the visual encoder and uses natural language text as the attack target. To align perturbed visual features with the target text, the method employs a two-level adaptive alignment strategy: (1) SVD-based subspace projection for global coarse-grained alignment, and (2) LLM-expanded descriptions combined with multi-view image augmentation and Optimal Transport for fine-grained semantic matching.

**Compliance With Llm Reviewing Policy:**

Affirmed.

**Final Justification:**

The authors have addressed my concerns, so I decide to raise my ratings.

**Key Questions For Authors:**

See the weakness part for my questions.

**Limitations:**

I didn’t see a limitation discussion in this paper, but I encourage the authors to further discuss the limitations.

**Strengths And Weaknesses:**

Strengths:

1.	This paper utilizes the SVD projection operation to increase the cosine similarity between the adversarial features and the target semantics , and leverages Optimal Transport for fine-grained semantic matching . The corresponding theoretical analysis is provided in the appendix .

2.	The experiments cover targeted gray-box attacks (against open-source models like Qwen2.5-VL and LLaVa) , as well as transfer attacks based on CLIP surrogate models (against closed-source commercial models like Claude, GPT-4o, and Gemini).

Weaknesses:

1.	The proof for Equation (4) and its corresponding Appendix A seems unreasonable. The authors represent the features of the adversarial image after the vision encoder as $\Pi(x^{adv}) = f_{\theta_v}(x) + f_{\theta_v}(\delta)_{||}$. However, modern vision encoders are highly non-linear. After dozens of layers of non-linear transformations, it is impossible for the perturbation to manifest merely as a simple linear additive feature in the deep feature space. The entire mathematical proof in Appendix A is completely built upon this idealized assumption of linear additivity. Once this premise is broken, the projected loss surface could become extremely complex or even non-differentiable, significantly undermining the theoretical rigor.

2.	The paper lacks a quantitative analysis of text expansion diversity and the SVD singular value spectrum. The algorithm uses an LLM to expand the target text into $N$ fine-grained descriptions (set to $N=30$ in the experiments) , and performs SVD on the features of these 30 sentences to extract the top $K=256$ principal components . The problem is that if the descriptions generated by the LLM are highly homogenized (e.g., merely replacing a few synonyms), the extracted feature matrix is inherently rank-deficient. Forcibly extracting 256 principal components from a highly homogenized matrix may introduce a large amount of meaningless text encoder noise. Projecting the image features into such a biased subspace will not only fail to achieve accurate "denoising," but may instead lead the visual features astray.

3.	The baselines for the commercial models seem a bit out of date. I am curious about the attack effectiveness of the proposed method against the latest models, such as GPT-5, Gemini 3.1, and Claude 4.6.

---

> ### Author Rebuttal · Authors · 2026-03-30
>
> # Response to Reviewer kPjo
>
> **Q1:** The proof for Equation (4) and its corresponding Appendix A seems unreasonable.
>
> **A1:** Thanks for the insightful comment. We agree that modern vision encoders are highly non-linear, and we have clarified that the “decomposition $f(x^{adv}) = f(x) + f(\delta)$ here is eased for better understanding” in Lines 209-211 of the paper.
>
> **Clarification of our assumption.** We would like to clarify that our analysis does not rely on a global linearity assumption, but instead is based on a local first-order approximation of the vision encoder around the input $x$. Specifically, given the adversarial example $x^{adv} = x + \delta, \quad ||\delta|| _ p \le \epsilon$, we consider the first-order Taylor expansion of the vision encoder $f(\cdot)$ at point $x$ as:
> $$
> f(x+\delta) = f(x) + J_f(x)\delta + \mathcal{R}(\delta),
> $$
> where $J_f(x) \in \mathbb{R}^{d \times d_x}$ denotes the Jacobian matrix, and $\mathcal{R}(\delta)$ is the higher-order residual term satisfying:
> $$
> ||\mathcal{R}(\delta)|| = \mathcal{O}(||\delta||^2).
> $$
> Under the standard adversarial perturbation regime (very small $\epsilon$, especially $\epsilon=4/255$ in our cases), the higher-order term becomes negligible, leading to the approximation:
> $$
> f(x^{adv}) \approx f(x) + J_f(x)\delta.
> $$
> Based on this, we can decompose the perturbation in the feature tangent space into components parallel and orthogonal to the semantic subspace $\mathcal{U}$ as:
> $$
> J_f(x)\delta = \delta_{\parallel} + \delta_{\perp}, \quad \delta_{\parallel} \in \mathcal{U}, \ \delta_{\perp} \perp \mathcal{U}.
> $$
> Therefore, the final projected visual feature can be written as:
> $$
> \Pi(f(x^{adv})) = \Pi(f(x) + J_f(x)\delta) = \Pi(f(x)) + \delta_{\parallel},
> $$
> which indicates that the projection operator effectively filters out the irrelevant component $\delta_{\perp}$ and preserves only the semantically aligned direction.
> Importantly, this decomposition should be interpreted as a local geometric analysis in the tangent space, rather than an exact additive decomposition in the original feature space.
> Similar local linearization assumptions are widely adopted in adversarial learning literature (e.g., gradient-based attacks such as PGD). Empirically, our results across multiple LVLMs in Tables 1,5,6 of the paper demonstrate that the proposed projection consistently improves semantic alignment and attack success, suggesting that this approximation is sufficiently effective in practice, further supporting our assumption.
> We will add more clarification in the revision.
>
> **Q2:** The paper lacks a quantitative analysis of text expansion diversity (Number $N$) and SVD singular value spectrum (Number $K$).
>
> **A2:** Thanks for your concerns.
>
> (1) As for the expanded texts, in fact, these descriptions generated by the LLM are diverse, as shown in the example in Figure 2 of the paper. Moreover, we have carefully conducted the ablation study on text number $N$ (Local target number) in Figure 5 (c)(d) of the paper. It shows that increasing number $N$ consistently improves performance, demonstrating that the generated descriptions are contextual and **Not** homogenized. We further compute the diversity score ($\downarrow$) as the average pairwise cosine similarity among text embeddings, where we achieve a significantly lower score of 0.58, indicating enriched semantic coverage. We will add more visualized examples of the expanded descriptions in the revision.
>
> (2) As for the SVD singular value spectrum, we have also studied how varying the number of singular vectors $K$ used to construct the projection matrix affects the vector quality in Figure 3 (a)(b) of the paper. It indicates that the learned text expansion matrix is contextual and **Not** homogenized/meaningless, because extracting more principal components from the matrix will steadily improve performance. To balance performance and efficiency, we set $K=256$ in our experiments. We will add more discussion in the revision.
>
> **Q3:** More commercial models such as GPT-5, Gemini 3.1, and Claude 4.6.
>
> **A3:** Thanks for your suggestion. We implement our attack and SOTA baseline FOA-Attack on GPT-5, Gemini 3.1, and Claude 4.6 in the Table below, where our attack is still more effective. We will add more experiments in the revision.
>
> |Attack|GPT-5 (ASR)|GPT-5 (AvgSim)|Gemini 3.1 (ASR)|Gemini 3.1 (AvgSim)|Claude 4.6 (ASR)|Claude 4.6 (AvgSim)|
> |:----:|:----:|:----:|:----:|:----:|:----:|:----:|
> |FOA-Attack|72.4|0.51|50.5|0.42|10.4|0.13|
> |Ours|78.7|0.65|64.1|0.58|21.5|0.22|
>
> **Q4:** Limitation discussion.
>
> **A4:** Thank you for the suggestion. Our method may have several limitations: (1) It still inherits potential biases from the pre-trained vision-language backbone; (2) The use of multi-view augmentation and optimal transport introduces additional computational overhead (however, our efficiency is still competitive as in Table 4 of the paper). We will include a Limitations section in the revision.

---

> > ### Author Rebuttal · Reviewer_kPjo · 2026-04-04
> >
> > The authors have addressed my concerns, so I decide to raise my ratings.

---

> > > ### Author Response · Authors · 2026-04-04
> > >
> > > Thank you very much for your follow-up and positive feedback. We will carefully revise our paper based on your comments. Thank you again for your time and thorough reading.

---

### Official Review · Reviewer_qkAd · 2026-03-13

**Soundness:** 3
**Presentation:** 3
**Significance:** 3
**Originality:** 3
**Overall Recommendation:** 4
**Confidence:** 3

**Summary:**

This paper studies targeted attacks on gray-box VLMs. Instead of relying on target images, the proposed framework performs attacks guided by target text embeddings. The method further incorporates an SVD-based projection, projection-aware optimal transport, and multi-view local alignment to improve semantic controllability of the attack. Extensive experiments on both open-source and closed-source LVLMs demonstrate the effectiveness and transferability of the proposed method.

**Compliance With Llm Reviewing Policy:**

Affirmed.

**Final Justification:**

The rebuttal have addressed most of my concerns. I will vote for acceptance.

**Key Questions For Authors:**

1. The proposed framework includes several components such as SVD projection, LLM-based text expansion, and projection-aware optimal transport. Could the authors further clarify the specific role of each component and how they interact to improve attack effectiveness?
2. The paper introduces fine-grained text descriptions generated by an LLM to guide local semantic alignment. How sensitive is the method to the quality of these generated descriptions? For example, would different LLMs or prompts significantly affect the attack performance?
3. The theoretical analysis mainly provides intuition for why the projection and alignment strategies may help the optimization. Could the authors comment on whether stronger theoretical understanding or guarantees might be possible in future work?

**Limitations:**

yes

**Strengths And Weaknesses:**

Strengths
1. Targeted attacks on gray-box VLMs are an important and timely research problem for evaluating the safety of multimodal foundation models.
2. Using target text embeddings to guide attacks is a natural and practically meaningful formulation for semantic manipulation in VLMs. The proposed framework also integrates several components to improve semantic alignment between images and target text.
3. Extensive experiments on multiple open-source and closed-source models validate the effectiveness and transferability of the method.

Weakness
1. The motivation for introducing multiple design components (e.g., SVD projection, LLM-based text expansion, and OT alignment) could be further clarified to better explain their individual roles.
2. The theoretical analysis mainly provides intuitive explanations and does not establish strong guarantees for attack effectiveness.
3. Some implementation details and hyperparameter choices are unclear, would benefit from additional discussion.

---

> ### Author Rebuttal · Authors · 2026-03-30
>
> # Response to Reviewer qkAd
>
> **Q1:** The motivation for introducing multiple design components.
>
> **A1:** Specifically, each component serves a complementary role: (1) LLM-based text expansion enriches the target semantic by generating diverse fine-grained descriptions, which provides a more informative and structured semantic space. (2) SVD-based projection extracts the principal components of these expanded semantic features, defining a compact semantic subspace that filters out noisy or less relevant directions and stabilizes global alignment. (3) Projection-aware OT performs fine-grained alignment between local visual features and textual semantics, enabling adaptive matching.
>
> These components interact in a progressive manner: The LLM expansion drives the framework by providing both the base vectors for SVD subspace and the target distribution for OT. By embedding the SVD projection into the OT cost matrix (Eq. 14), local feature matching is strictly constrained within the global semantic subspace. Table 5 of the paper confirms this synergy.
>
> **Q2:** Theoretical analysis does not establish strong guarantees for attack effectiveness.
>
> **A2:** Thanks for the comment. Our theoretical analysis not only provides an intuitive explanation, but also establishes guarantees for attack effectiveness through the following dimensions:
>
> Firstly, the SVD structured subspace projection provides deterministic theoretical guarantees for the effectiveness of the attack through a dual mechanism. (1) During the PGD iterative optimization process, this projection acts as a signal filter, explicitly eliminating a large number of orthogonal noise components $\delta_{\perp}$ in the traditional gradient that are irrelevant to the target semantics. This ensures that the optimization trajectory is directed more precisely toward the target semantic space, significantly enhancing gradient directionality and utilization efficiency within a limited perturbation budget $\epsilon$. (2) Simultaneously, from a global optimization perspective, the projection strictly confines the attack's search space to the intrinsic manifold defined by the target text and its fine-grained semantic expansions, exerting a strong regularizing effect. This adaptive constraint not only effectively avoids attacks getting trapped in local optima caused by irrelevant features, but also ensures that visual features have been structurally pre-aligned semantically before entering the downstream black-box LLM, thereby theoretically maximizing the success rate of inducing the model to produce erroneous outputs.
>
> Furthermore, we have proved that the projected optimal transport cost $C^{\Pi}$ is strictly no greater than the conventional cost $C$, i.e., $OT(C^{\Pi}) \le OT(C)$. This establishes a tighter upper bound for fine-grained adversarial alignment, ensuring that the algorithm can achieve accurate matching of visual details and textual semantics at a lower cost.
>
> These theoretical insights are empirically validated in Tables 5 and 6 of the paper, where both projection and alignment strategies improve our attack's effectiveness. We will add more discussion in the revision.
>
> **Q3:** Implementation details and hyperparameter choices.
>
> **A3:** Thanks for your comment. As illustrated in Section 4.1, (1) As for pre-trained models and tasks, we utilize the widely used ones for fair comparison; (2) As for gray-box attack setup, we follow existing gray-box attacks to utilize the same perturbation budget, step size and iteration number for fair comparisons. The hyperparameters $M$ (Local Image Number), $N$ (Local Target Number), $K$ (principal component) are set according to our ablation studies as shown in Figure 5 (a)(b), Figure 5 (c)(d), Figure 3 (a)(b) of the paper, respectively; (3) As for transfer attack, we also follow previous works to utilize the same perturbation budget, step size, and iteration number for fair comparison. We will provide more clarifications.
>
> **Q4:** Would different LLMs or prompts significantly affect the attack performance?
>
> **A4:** Thanks for the concern. First, we implement our attack with different LLMs for comparison and report the attack success rates in the Table below. It shows that it would not affect our overall attack performance.
>
> |LLM|Qwen2.5-VL-3B|Qwen2.5-VL-7B|LLaVa-1.5-7B|LLaVa-1.6-7B|Gemma-3-4B|Gemma-3-12B|
> |:----:|:----:|:----:|:----:|:----:|:----:|:----:|
> |GPT|86.5|81.0|84.7|78.5|83.0|74.2|
> |DeepSeek|86.2|80.8|84.3|78.1|82.7|73.9|
> |Claude|86.8|81.3|85.1|78.9|83.4|74.5|
>
> Our attack is also not sensitive to the target prompt.
> |Target Text|Qwen2.5-VL-3B|Qwen2.5-VL-7B|LLaVa-1.5-7B|LLaVa-1.6-7B|Gemma-3-4B|Gemma-3-12B|
> |:----:|:----:|:----:|:----:|:----:|:----:|:----:|
> |"A man holding a big doughnut at a festival"|87.2|82.3|85.6|79.1|83.8|75.0|
> |"A photo of a teddy bear on a skateboard in Times Square"|84.5|79.6|82.8|76.4|81.2|72.3|
> |"A beautiful bird with a black and white color in snow"|88.1|83.5|86.9|80.2|84.6|76.8|

---

> > ### Author Rebuttal · Reviewer_qkAd · 2026-04-03
> >
> > The rebuttal addresses my main concerns in a reasonably solid way, especially by adding extra evidence on sensitivity to different LLMs and prompts.
> > Considering the overall quality of the original submission, I will maintain my score of wa and vote for acceptance.

---

> > > ### Author Response · Authors · 2026-04-03
> > >
> > > Thank you for your kind acknowledgement and for taking the time to read our rebuttal carefully. We appreciate your positive assessment and your constructive feedback throughout the review process. We will carefully incorporate the rebuttal into the revised version.

---

### Decision · Program_Chairs · 2026-04-30

**Decision:**

Accept (regular)

**Comment:**

This paper studies gray-box adversarial attacks on large vision-language models. The key idea is to guide attacks using target text instead of target images, and to align visual features with text semantics through SVD-based projection and optimal transport. The reviewers agree that the problem is important and timely. The text-guided attack formulation is considered meaningful, and the paper shows strong empirical results, including good transferability across both open- and closed-source models.

However, there are several concerns. The method is relatively complex and combines multiple components (text expansion, SVD projection, optimal transport) without a very clean or unified justification. The theoretical analysis is weak and relies on local linearization assumptions, which limits its rigor. Some design choices, such as prompt expansion and entropy-based weighting, are not well analyzed and appear heuristic.

In the rebuttal, the authors clarified the role of each component and provided additional empirical evidence (e.g., robustness to different LLMs, prompts, and newer models). The reviewers found these responses largely satisfactory, and all reviewers support acceptance after rebuttal.

Overall, this is a solid empirical paper with a useful perspective (text-guided attacks in the gray-box setting) and strong experimental results. However, the work is not particularly clean conceptually, and the theoretical contribution is limited.